# Impact of G-Quadruplexes and Chronic Inflammation on Genome Instability: Additive Effects during Carcinogenesis

**DOI:** 10.3390/genes12111779

**Published:** 2021-11-09

**Authors:** MaryElizabeth Stein, Kristin A. Eckert

**Affiliations:** Department of Pathology, The Jake Gittlen Laboratories for Cancer Research, Hershey, PA 17033, USA; mss82@psu.edu

**Keywords:** G-quadruplex, oxidative stress, genome instability, DNA polymerases

## Abstract

Genome instability is an enabling characteristic of cancer, essential for cancer cell evolution. Hotspots of genome instability, from small-scale point mutations to large-scale structural variants, are associated with sequences that potentially form non-B DNA structures. G-quadruplex (G4) forming motifs are enriched at structural variant endpoints in cancer genomes. Chronic inflammation is a physiological state underlying cancer development, and oxidative DNA damage is commonly invoked to explain how inflammation promotes genome instability. We summarize where G4s and oxidative stress overlap, with a focus on DNA replication. Guanine has low ionization potential, making G4s vulnerable to oxidative damage. Impacts to G4 structure are dependent upon lesion type, location, and G4 conformation. Occasionally, G4s pose a challenge to replicative DNA polymerases, requiring specialized DNA polymerases to maintain genome stability. Therefore, chronic inflammation creates a dual challenge for DNA polymerases to maintain genome stability: faithful G4 synthesis and bypassing unrepaired oxidative lesions. Inflammation is also accompanied by global transcriptome changes that may impact mutagenesis. Several studies suggest a regulatory role for G4s within cancer- and inflammatory-related gene promoters. We discuss the extent to which inflammation could influence gene regulation by G4s, thereby impacting genome instability, and highlight key areas for new investigation.

## 1. Introduction

In their seminal 2000 paper, Hanahan and Weinberg proposed the hallmarks of cancer, functional changes that are acquired during the multistep development of all human tumors. Later, the authors proposed that acquiring these hallmarks is made possible by the two “enabling characteristics” of genome instability and inflammation [1]. Importantly, the enabling characteristics of genome instability and inflammation are intricately interconnected at the mechanistic level [2,3].

### 1.1. Non-B DNA Structure Formation Drives Some Forms of Genome Instability

Genome preservation necessarily relies on accurate and efficient DNA synthesis and functional DNA repair processes. Although lesions from exogenous or endogenous sources and/or defective repair contribute to genome instability, another characteristic of DNA must also be considered, namely, primary DNA sequence and repetitive sequences. Repetitive sequences can form a variety of DNA secondary structures (collectively called non-B DNA structures) that induce variable chromosomal alterations in the human genome [4]. In fact, secondary structure can be a considerable contributor to genome instability, when compared to the primary DNA sequence itself [5]. A detailed analysis of nucleotide variant frequency from the 1000 Genomes Project showed a substantial enrichment of variants at sites of potential non-B DNA, compared to B-DNA [6]. Likewise, analysis of whole genome sequencing of cancer genomes revealed a correlation between non-B DNA forming sequences and somatic mutations [7]. 

Outside of inducing genome instability, non-B DNA may contribute to changes in gene and protein expression, depending on location [5]. Certain non-B DNA types (i.e., slipped DNA, G-quadruplex, Z-DNA) are particularly prominent at or near transcriptional start sites [6], suggesting a location-dependent function of structured DNA. Analyses of expression quantitative trait loci (eQTL) showed reduced variants at sites of non-B DNA [6]. Taken together, these studies, as well as others not mentioned here, show different ways in which non-B DNA can contribute to genome instability. For recent, detailed reviews on repetitive DNA sequences and non-B DNA structures, see Khristich and Mirkin [8], and Poggi and Richard [9]. 

### 1.2. Chronic Inflammation and Tumor Cell Evolution

Chronic inflammation is a key physiological mediator of cancer development, promoting the outgrowth of mutant cells and enabling malignant progression. Inflammation causes increased mucosal production of pro-inflammatory cytokines and dysregulated redox signaling within the tissue microenvironment which contribute directly to malignant progression [10,11,12]. Insights into the mechanistic relationship between inflammation and genome instability have been gained by studies of chronic inflammatory conditions, such as ulcerative colitis (UC). UC patients have an increased lifetime risk of colorectal cancer, and the progression to colitis-associated cancer has been well studied in both animal models and human tissues (reviewed in [13,14,15]). Based on this research, a well-accepted model proposes that inflammation-induced DNA damage in epithelial cells promotes genome instability. Reactive oxygen and nitrogen species (RONS) generated during the inflammatory response result in direct DNA, lipid, and protein damage to colon epithelial cells [16,17]. The impact of inflammation-associated DNA damage on tumor development has been tested directly using a dextran sulfate sodium (DSS)-induced mouse model of chronic inflammatory bowel disease. RONS-induced base lesions are found in both acute and chronic DSS-treated mice, and lesion levels are increased in colonic mucosa and tumors of base excision repair-deficient mice treated with DSS [18,19]. Many types of genome instability have been detected throughout phenotypically normal and pre-malignant colon tissues of UC patients, including base substitutions [17], microsatellite instability [20,21], large deletions [22], and chromosomal instability [23]. A recent whole genome sequencing analysis of non-neoplastic UC patient colon samples revealed an overall 2.4-fold increase in mutation rate compared to control colons. Consistent with earlier studies, the mutational burden included excess base substitutions, indels, and structural variants [24].

Elucidating the interplay between DNA sequence and chronic inflammation is needed to reveal the full spectrum of genome instability mechanisms operating in pre-malignant tissues under chronic inflammation. Here, we discuss the replication of a specific type of non-B DNA structure, the G-quadruplex, in the context of oxidative damage and consequences for genome instability (Figure 1).

## 2. G-Quadruplex: A Non-B DNA Structure That Contributes to Genome Instability 

### 2.1. G-Quadruplexes (G4s): Sequence and Structure Heterogeneity

The ability of G-rich sequences to fold into four-stranded structures was demonstrated biochemically in 1988. Sen and Gilbert showed that G-rich sequences containing four separate G-tracts of at least three contiguous guanine bases each form both intra- and intermolecular four-stranded structures [25,26]. These structures, known as G-quadruplexes (G4s), are formed by Hoogsteen base pairing of the guanines from these G-tracts, separated by N loops (G_≥3_N_≥1_G_≥3_N_≥1_G_≥3_N_≥1_G_≥3_, where N is any nucleotide), and stabilized by a monovalent cation [27]. Though beyond the scope of this review, it is important to note that G4 motifs are conserved in the genomes of multiple species, spanning across kingdoms and phyla. While all functions of G4s have not been fully elucidated, many G4 motifs in the human genome and genomes of other species are located in regulatory regions [28,29,30] and can be considered to be functional [31]. 

G4s form several different conformations (parallel, antiparallel, and hybrid) and are either intramolecular, bimolecular, or tetramolecular in nature [32]. Cations are critical to structure formation and stability. Ionic radius, hydration energy, and guanine oxygen coordination are properties that determine which cations favor G4 stabilization or destabilization, and these have been validated through biophysical and biochemical experiments [32,33,34]. K^+^ and Na^+^ are physiologically relevant monovalent cations most favorable for forming and sustaining G4 formation and have been used extensively in vitro to promote G4s. Between the two, K^+^ is most favorable due to a decrease in electrostatic repulsion of K^+^ ions located between tetrads, compared to Na^+^ ions located within tetrads [34]. These properties suggest that regulation of G4s may be, in part, reliant on the type of cation and cation concentration present in vivo (i.e., in cells or tissues) [32]. 

Approaches to demonstrating G4 formation in vivo rely on detection by G4 structure-specific antibodies. For example, the G4-specific BG4 antibody displays high affinity to both intra- and intermolecular G4 structures, without a preference for the type of G4 conformation, and with no binding to ssDNA or dsDNA [35]. A more recent study details a live cell imaging method using a fluorescence probe, DAOTA-M2, to overcome challenges with other approaches and antibody use. This method relies on the fluorescence lifetime of the probe, dependent on whether it is bound to dsDNA, RNA, or G4s, and has shown promise for use in different cell lines [36]. 

Another widely used experimental approach to infer the effects of G4 formation relies on G4-stabilizing drugs. A few stabilizers (e.g., telomestatin, BRACO19) have been characterized for binding to the telomeric G4 using molecular dynamics simulations. These simulations showed stabilizer binding is defined largely by a combination of van der Waals, hydrophobic, and electrostatic interactions [37]. Pyridostatin (PDS) is one of the more widely used G4 stabilizers. Circular dichroism and ultraviolet resonance Raman spectroscopy data show the ability of PDS to bind to more than one type of G4, albeit the strongest interaction occurs with a parallel G4, and the weakest interaction is with a hybrid G4. Structural properties of PDS also suggest variable binding behaviors, with preferential interactions between PDS and G4 loop adenine and thymine bases [38]. Treatment of cells with low concentrations of PDS increases double-strand breaks, as visualized by γH2AX foci, but these loci do not colocalize with the telomere binding protein TRF1, supporting the occurrence of G4s outside of telomeres [39]. The use of G4 stabilizers in experiments increases our understanding of the cellular occurrence of these structures. Understanding the G4 stabilizer properties necessary for interacting with different types of structures will be essential as the field designs and improves G4 stabilizers for therapeutic use.

### 2.2. G4s Contribute to Genome Instability

The mutagenic potential of G4 sequences was elegantly described by the Tijsterman lab, using a *Caenorhabditis elegans (C. elegans)* genetic model system [40,41,42]. These authors showed that endogenous G4-forming motifs are highly mutagenic in vivo in animals that lack *dog-1*, *the C. elegans* ortholog of mammalian FANCJ helicase. Deletion frequency was enhanced in G4-containing sequences, with most deletions ~50–300bp in length [40,41]. Deletions were primarily observed at the 3′ flanking sequence of a G4, which indirectly suggested to the authors the occurrence of polymerase stalling [41,42]. These experimental studies highlight that G4s can be hotspots for mutagenesis *in vivo,* and the influence of G4s on the flanking sequences. 

Genome instability caused by G4 formation within human minisatellite repeats has been investigated using an *Saccharomyces cerevisiae (S. cerevisiae)* model system. The highly polymorphic, G-C rich CEB1 minisatellite is comprised of a 39 nucleotide repeat unit that forms a well-characterized G4 structure. A 42 unit CEB1 array inserted downstream of the ARS 305 origin is relatively stable in wild-type yeast strains; however, when cells are grown in the presence of the G4 stabilizer Phen-DC_3,_ CEB1 instability (measured as expansions and contractions of the repeats) is observed [43]. In contrast, the CEB25 minisatellite, which forms a similar parallel G4 conformation, is stable in the same assay. Extensive mutagenesis of the CEB25 motif revealed a pronounced impact of loop length, base composition, and position on G4-induced genome instability [44]. Specifically, G4 motifs with short (≤4 nt) loops of pyrimidines display high thermostability and instability. Importantly, both the CEB1 and CEB25 tandem arrays stimulate the rate of gross chromosomal rearrangements in *S. cerevisiae* in a manner consistent with G4 formation [45].

Several groups have examined the genome-wide association of G4 motifs with sites of increased genome instability in cancer genomes. This approach utilizes computer algorithms to predict DNA sequences with G4-forming potential (PG4, potential G4). Although the precise tools vary from study to study, most computational approaches are based on rules derived from biochemical/biophysical characterization of a few sequences (see [46] for a recent review comparing G4 computational methods). Across cancer genomes (pan-cancer), hotspots of somatic copy number variant breakpoints are significantly enriched for PG4s, along with other repeated sequences [47]. Similarly, analyses of translocation and deletion endpoints in the COSMIC database revealed a significant association with PG4 motifs [48]. In addition, a recent analysis of cancer genomes demonstrated significant enrichment of base substitutions and insertion-deletion mutations near PG4s, across tumor types [7].

### 2.3. Relationship of G4 Formation to DNA Replication

Current experimental evidence supports the hypothesis that G4 structures are formed during DNA replication. In MCF7 cells, BG4 foci are highest in S phase cells and decrease in number upon treatment with aphidicolin (a replication stress inducer that inhibits replicative polymerases [49]), consistent with G4 structure formation during DNA replication [35]. U2OS cells display distinct nuclear BG4 antibody foci dispersed across chromosomes, primarily outside of telomeres, and foci are increased upon PDS treatment, consistent with a genome-wide distribution of G4 structures [35]. These data support computational studies that have suggested more than 350,000 PG4s reside outside of telomeric regions; with the inclusion of non-canonical motifs, the total number of PG4 motifs is over 700,000 in the human genome [50,51]. Recently, single-molecule fluorescence microscopy was combined with unbiased pattern recognition algorithms to analyze G4 structures associated with replication [52]. In U2OS cells, ~2% of replisomes, identified by PCNA and MCM helicase antibodies, colocalize with G4 structures.

Other evidence demonstrates an intimate relationship between G4 structures and replication. Origin G-rich repeated elements (OGREs) are functionally associated with sites of DNA replication initiation in mouse cells [53]. OGREs containing biophysically confirmed G4-forming motifs can function as origins of replication when integrated at ectopic genome locations, and such sequences can inhibit nuclear DNA replication in Xenopus extracts. In human cells, the majority (83%) of core origins identified genome-wide by short nascent strand sequencing of multiple cell lines contain at least one PG4 motif within a G-rich sequence element [54]. 

Though OGREs point to a positive functional impact of G4s, several studies suggest G4s negatively impact replication. Expanded G-rich microsatellites associated with neurodegenerative diseases, such as Fragile X-syndrome (e.g., [CGG]_n_ and [CCG]_n_), cause replication fork stalling in vivo [8]. While these sequences can form several non-B DNA structures, the Usdin lab used biochemical assays with purified proteins to show that the formation of four-stranded structures directly arrests DNA polymerases in vitro [55,56]. The well-known “polymerase stop assay” demonstrated that prokaryotic DNA polymerases arrest synthesis when utilizing poly (G) templates with four tracts of four or more consecutive guanine bases. The arrest occurs immediately preceding the 3′ G-tract, is K^+^-dependent, and is not observed on the complementary C-rich strand. These data provide compelling evidence that the formation of intramolecular G4 structures in the ssDNA template preceding the DNA polymerases act as a barrier to DNA synthesis. These data also form the basis for a G4-Seq approach to map PG4s across the human genome [51]. DNA polymerase inhibition during Illumina sequencing through PG4 motifs results in significantly decreased sequencing quality (Phred) scores. By comparing quality scores after sequencing human cell DNA in the absence or presence of KCl or PDS, Chambers et al. mapped ~380,000 PG4 motifs genome-wide, including those at the experimentally verified *c-myc* and *c-kit* genes. Importantly, 70% of the PG4s are non-canonical motifs, comprised of either G tracts with long loops (>7 bases) or with one interrupted G tract [51]. Because all these PG4 motifs likely have varying stabilities, it can be assumed that PG4s will have variable effects on genome-wide replication. 

In the *C. elegans* mutagenesis studies described above, G4-induced genome instability is inferred to be caused by a blocked replication fork. The *S. cerevisiae*/CEB genome instability model (above) also has been used to investigate the effects of replication on instability [43]. The CEB1 minisatellite array was chromosomally inserted in two orientations, such that the G4 motif was present on either the leading strand or lagging strand of replication forks emanating from ARS305. CEB1 instability was greatly increased in Pif1-deficient cells only when the G4 motifs were present on the leading strand. Two-dimensional gel electrophoresis was used to examine fork progression through the CEB1/G4 region. No fork pausing was observed in wild-type cells, whereas aberrant replication intermediates (X-spikes) were observed in Pif1-deficient cells, but only when the G4 motif was present on the leading strand. Mutations that abolish G4 structure formation also abolished aberrant replication and decreased instability in the Pif1-deficient cells [43]. 

Another experimental system, using DT40 chicken cells, has reached a similar inference; namely, that some G4s can block replication fork progression (for a recent review, [57]). This approach monitors epigenetic instability caused by G4 motifs, which is inferred to result from G4-induced replication fork pausing and the creation of a gap in newly replicated DNA after fork progression resumes. Because post-replication gap-filling is not coupled to proper histone recycling, fork pausing/restart results in altered epigenetic marks in the vicinity of the G4 secondary structure [58]. If the G4 motif is within a gene, this process can be measured in a cell population by heritable changes in expression, and therefore is an indirect measure of replication perturbations. This experimental model has been used recently to demonstrate a role in G4 replication for the replication fork protection complex proteins, Timeless and Tipin [59]. Deletion of either Timeless or Tipin in DT40 cells increases G4-induced epigenetic instability. Moreover, Timeless binds to G4s with high affinity. Timeless also interacts with DDX11, a structure-specific helicase, and deletion of DDX11 in the DT40 model induces G4-dependent epigenetic instability.

Together, the current in vivo studies, although indirect, are consistent with perturbed replication fork progression through some G4 structures. Clearly, not all G4 motifs have a negative impact on replication. Further investigation with more direct measurements of replication fork pausing/arrest is needed to elucidate all the parameters determining which G4s are detrimental to replication. In addition, the mechanism underlying G4-induced fork pausing remains to be determined. Although inhibition of replicative polymerases is a popular model (evidence for this is discussed below), other mechanisms should be considered. For instance, several proteins associated with replication bind with high affinity to G4 structures, and bound proteins are also obstacles to replication fork progression. In addition, many fork-associated helicases (e.g., FANCJ) show in vitro activity towards G4 structures, and have been implicated in G4 replication. For reviews concerning these structure-specific DNA helicases and the evidence that these act to prevent replication fork stalling at G4 motifs, see [57,60,61].

### 2.4. DNA Polymerases Implicated in G4 Maintenance

How do cells maintain replication through G4 motifs under physiologically normal conditions? Below, we summarize the evidence regarding DNA polymerase synthesis through G4 motifs. The specialized Y-family polymerases (e.g., pol eta (η) and kappa (κ), Rev1) have been of particular interest in this regard. 

#### 2.4.1. Replicative DNA Polymerases 

Studies investigating the impact of G4 structure formation on DNA synthesis in vitro by eukaryotic replicative DNA polymerases (pols) have shown that some G4s can negatively impact synthesis efficiency. Synthesis by human and yeast pol delta (δ) holoenzymes (in the presence of accessory proteins) is stalled when using telomeric repeat templates, but only under certain conditions (short repeats and linear substrates) [62]. Importantly, this study showed that Pol δ synthesis is not blocked by G4 formation within long telomeric repeats, raising the possibility that dynamic G4 structure formation may allow polymerases to synthesize through the repeats without stalling. Another study confirmed that the yeast Pol δ holoenzyme pauses transiently within a telomeric G4 repeat; however, Pol δ synthesis is inhibited at the base of the more stable *c-myc* G4 sequence [63]. The biophysical features of G4 motifs needed for Pol δ inhibition were found to be related to short loop lengths, which increase thermodynamic stability, and parallel G4 conformation [63]. The catalytic core of human pol epsilon (ε) was also not efficient at synthesizing templates containing the *c-myc* promoter G4 motif [64]. Clearly, more experimental evidence is needed to fully elucidate the full range of G4 structure effects on eukaryotic replicative DNA polymerases. 

#### 2.4.2. Rev1

In DT40 chicken cells, the absence of the dCTP terminal transferase Rev1 disrupts histone recycling and increases G4-induced epigenetic instability [58]. These data suggest a potential role for Rev1, either directly or indirectly, in G4 maintenance, possibly post-replicative gap-filling synthesis. Purified Rev1 can bind parallel G4s in vitro [65,66] and interrupt G4 formation [65]. In Rev1 knockout HAP1 cells, G4-containing plasmids display substantially increased mutation frequencies in the presence of PDS, while Rev1-complemented knockout cells display reduced mutation frequency at G4s [66], suggesting that Rev1 plays a role in sustaining low error rates during G4 synthesis. 

#### 2.4.3. Y Family Polymerases

Specialized polymerases η and κ are reasonable alternatives for the cell to utilize in sustaining replication at G4s. However, direct evidence to support roles for pols η and κ in synthesizing G4s remains elusive. Eddy et al. showed that the catalytic cores of both human pols η and κ efficiently bind G4s and have enhanced fidelity at G4 motifs in vitro [64,67]. However, a study by Edwards et al. observed some stalling of η and κ at telomeric G4s when stabilized by lower temperatures. Nucleotide misincorporation and extension by these polymerases was also observed when increasing temperature relaxed the G4 [68]. Though Edwards et al. suggested pols η and κ might not be critical in replicating G4s, it is not immediately clear if temperature and telomeric G4 conformation and/or sequence has an impact on impact polymerase activity, compared to the parallel G4 investigated by Eddy, et al. [64,67]. Interestingly, in the presence of KCl, the catalytic core of pol η stalled at the G4 in vitro, compared to its activity in the presence of NaCl [64]. Because the presence of K^+^ more favorably promotes G4 formation over Na^+^, a less stable G4 may be formed in NaCl, allowing for flexible interaction between the catalytic core of η and the G4 in this context. Though these conditions are physiologically relevant, it is difficult to extrapolate these results to cells, especially when only the catalytic domain of the polymerases was studied. 

Only a few ex vivo studies have examined the roles of Y family polymerases. Betous, et al. observed sensitization to telomestatin in U2OS cells with pol η or κ knockdown, suggesting the presence of these specialized polymerases is important for overcoming G4 stabilization, but neither compensates for the other [69]. Increased DNA damage, determined by the presence of γH2AX, was observed in pol η or κ depleted HeLa cells containing plasmids harboring the c-*MYC* promoter region containing a G4 motif and two other types of non-B DNA sequences, compared to HeLa cells with B-DNA plasmids [69]. Overall, this study suggests that pol η and κ may have some role in preventing non-B DNA-induced damage. It will be important to determine if these results are observed in other cell lines, and in what manner pols η and κ may be recruited to sites of G4s. Taken together, the available evidence suggests Y-family polymerases may function at sites of G4s. Possible recruitment of Rev1 to disrupt G4 formation [58] and act as a scaffold for pols η and κ, or recruitment of a specialized polymerase like pol η to take over at the replication fork when a G4 is encountered [64] are plausible hypotheses that warrant further investigation. 

#### 2.4.4. POLQ

Pol theta (θ; *POLQ* gene) has been shown to maintain translesion synthesis (TLS) and suppress skin tumorigenesis. Yoon, et al. demonstrated that in the absence of pol η, pol θ facilitates TLS of UV-induced lesions [70]. Pol θ^−/−^ pol η^−/−^ mice display increased skin tumor incidence, compared to pol η^−/−^ mice, indicating pol θ exerts a protective effect in repressing skin tumorigenesis. Additionally, elevated sister chromatid exchanges were observed in pol θ-deficient mouse cells, independent of UV irradiation. In both human and mouse embryonic fibroblasts depleted for pol η, fork progression and cell survival were decreased after pol θ depletion, suggesting pol θ compensates for a loss of pol η [70]. 

The role of pol θ in maintaining genome integrity is not restricted to UV-induced lesions. In the *C. elegans* studies described above, deletion events are *polq*-dependent [41,42]. Pol θ contributes to the repair of DNA breaks caused by failed replication through G4-motifs via theta-mediated end joining (TMEJ), a conclusion supported by the specificity of deletion events [41]. TMEJ could represent a compensatory mechanism, active when other pathways of G4 synthesis are not possible to maintain genome integrity. Therefore, it will be important to determine if TMEJ is more frequently induced in cells that are deficient in another specialized polymerase. 

Indeed, a more recent study highlights a need to investigate *POLQ* in this context ex vivo. A CRISPR-Cas9 dropout screen to assess cellular sensitivity associated with G4 stabilization was conducted using HCT116 cells treated with CX-5461, PDS, or BMH-21 (a non-G4 binding RNA pol I inhibitor) [71]. CX-5461 is an RNA pol I inhibitor and G4 stabilizer, currently in two phase I clinical trials for solid tumors (NCT02719977, NCT04890613). Several genes related to DNA replication, damage response, and chromatin remodeling, among others, were identified in this screen, including *POLQ*. Sensitivity to CX-5461, but not BMH-21, was validated in multiple cell lines with sgRNA or siRNA-mediated *POLQ* deficiency [71], further supporting a possible role for pol θ maintaining genome stability at G4 motifs. 

#### 2.4.5. PrimPol

The polymerase-primase enzyme PrimPol also has been implicated in replication fork progression through G4 regions. The catalytic and zinc finger domains of PrimPol are critical to its activity at G4s, as evidenced using the DT40 epigenetic assay described above [72]. Though PrimPol cannot synthesize DNA past G4s in vitro [72,73], it can bind with high affinity to stable G4s [72], much like Rev1 and specialized pols η and κ. Addition of replication factors such as PCNA or RPA did not improve PrimPol’s synthesizing capability at G4s [72]. However, the primase activity of PrimPol can reprime the template, downstream of the G4, allowing synthesis to continue in vitro. These data support PrimPol as an active player in overcoming the G4 barrier during replication. Further studies are necessary to determine how, and in what cellular context, PrimPol might be recruited when replicating G4s. 

## 3. Direct Impact of Chronic Inflammation: G4 DNA Oxidation

Oxidative lesions in DNA can arise in response to exogenous or endogenous sources that generate RONS, including inflammation (Section 1.2). These RONS remove an electron from a DNA base, creating an electron hole that gets transferred to bases of lower ionization potential [74,75]. Guanine possesses the lowest ionization energy of all four bases, making it highly susceptible to oxidative damage [74,75,76]. Left unrepaired, these DNA lesions can lead to base substitutions. Indeed, computational analysis has indicated an association between base substitutions and electron transfer, highlighted by ionization energy of the nucleotides and base stacking interactions [74]. Here, we review the impacts of guanine oxidation in G4s, with a brief look at possible impacts of oxidation on other bases located in loop sequences. For a detailed discussion on oxidative lesions and their intermediates, see a review by Cadet and Wagner [76]. 

### 3.1. Guanine Oxidation at G4 Sequences

#### 3.1.1. Types of Lesions

Given the low ionization potential at guanines, it is easy to see the problem that oxidation poses to non-B DNA with consecutive G-tracts. An analysis of variants within mononucleotide repeats, using data from the 1000 Genomes Project, revealed that base substitutions occurring at G-tracts are associated with electron transfer rather than template misalignment or slippage [77]. Given what we know about guanine oxidation, this correlation may be a good predictor of the impact of chronic oxidation on DNA in cells. One of the most notable and well-studied oxidative lesions is 8-oxo-7,8-dihydroguanine (8oxoG), which results from the interaction of guanine with a hydroxyl radical. The 8oxoG can base pair with adenine, subsequently resulting in a G > T transversion during replication [78]. This consequence may be more substantial at sequences containing consecutive guanine bases.

Further oxidation of 8oxoG can occur, producing products such as spiroiminodihydantoin (Sp) and 2,6-diamino-4-hydroxy-5-formamidopyrimidine (FapyG). In vitro comparisons of purified telomeric G4 oligodeoxynucleotides showed that the major oxidation products differ between guanines within duplex DNA and those within G4s [79]. Interestingly, guanine oxidation in duplex DNA yielded different products dependent on the oxidant, but guanine oxidation in G4s yielded Sp as the major product [79]. Whether this difference in oxidation products is unique to telomeric G4s has yet to be determined.

#### 3.1.2. Dependence on Location and G4 Conformation

Though the mutagenic potential of oxidation products, such as 8oxoG, Sp, and FapyG, likely occurs at G4 motifs in cells under oxidative stress, we also need to assess other impacts of this microenvironment on G4 stability. In a state of oxidative stress, are all guanines within a G4 susceptible to potential damage? This interesting question may seem simple on its surface, but in fact, is complex. In terms of electron transfer within mononucleotide repeats, simulations highlight a dependence of where the substitution occurs on G-tract length [77]. In a shorter G-tract, the initial position may be most frequently targeted by oxidation, as predicted by increased mutation rates at that position, but the longer the G-tract, the more likely electron transfer will occur, as indicated by a shift in mutations downstream of the initial G [77]. For G4s, oxidation at the most 5′ guanines is dominant; however, the formed secondary structure appears to determine location of oxidative lesions. In vitro, one-electron oxidants (e.g., riboflavin) most frequently acted on guanines that were part of the external tetrads of the G4 (i.e., the most 5′ Gs of each G-tract), while other types of oxidants, such as H_2_O_2_ and Cu, did not show a preference for guanine location [79]. K^+^ or Na^+^ are favorable in maintaining G4 formation, and steric hindrance and charges associated with different lesions would necessarily interrupt the metal cation coordination. Miclot, et al. observed instances of cation expulsion in a telomeric G4 via molecular dynamics simulations, particularly when two 8oxoG lesions were present [80], suggesting oxidative lesions may impact cation arrangement and coordination. However, the telomeric G4 motif containing 8oxoG could still bond with other nucleotides, resulting in rearrangement of base conformations that have minor impacts on structure in vitro. In fact, biophysical measurements showed no change in the parallel structure and only slightly reduced peaks in the hybrid structure upon H_2_O_2_ addition, suggesting that this oxidant does not create lesions that change the overall G4 structure [80].

Dependence on location is a predominant theme that is not limited to telomeric G4s. Effects of 8oxoG were evaluated recently for the G4 antiparallel structure within the P1 promoter of *Bcl2* [81]. NMR and UV melting demonstrated that 8oxoG at guanines located in an external tetrad and loop did not show significant impacts on G4 structure, whereas 8oxoG located at middle tetrad positions resulted in decreased thermostability and broader imino resonance peaks, consistent with an effect of 8oxoG on G4 structure. This might be due to formation of an antiparallel structure with base orientations (i.e., *anti* or *syn*) and bonding partners that differ from the non-damaged sequence [81]. Though the entire G-rich sequence in the *Bcl2* promoter was not evaluated, this study points to an interesting phenomenon that could translate to cells under oxidative stress. Minimal effects of oxidative damage at certain guanines could be observed for other G4s in regulatory regions. As a result, G4 stability could be retained such that oxidative damage does not pose major functional consequences. 

Another study demonstrated that the degree of interruption and unfolding of the telomeric G4 structure by oxidation is dependent on the type of lesion and location of the guanine affected in vitro, with the central guanine modifications causing the most destabilization [82]. Telomerase activity has been investigated extensively in the context of oxidation. Much like G4 stability, the effects on telomerase are dependent on the location of the lesion within the G4 [83,84] and dependent on whether the lesion occurs in the dNTP pool or in the motif itself [83]. Extension by telomerase in vitro is impeded when an 8oxoG is inserted from the dNTP pool [83] or when an oxidative lesion occurs within the terminal G-tetrad at the end of the sequence [82]. However, at other positions, oxidative lesions do not inhibit telomerase binding and extension, possibly due to minimal effects to base pairing on the RNA template compared to the effects of point mutations [82]. 

An outstanding question is whether adaptability of G4 structures to oxidative base lesions is dependent on surrounding nucleotides, loop sequences, or original conformation (parallel, antiparallel, and hybrid). Indeed, the consequences of oxidative damage at G4-forming genome sequences in cells have only recently been investigated. The Opresko lab developed an experimental tool to target 8-oxoG damage specifically to telomeres. Using this approach, the lab revealed that chronic 8oxoG formation and persistence at telomeres impairs replication, resulting in telomere shortening and genome instability [85]. An immunofluorescence study showed that non-tumorigenic MCF-10A breast epithelial cells express elevated G4 levels that colocalize with 8oxoG upon H_2_O_2_ treatment [80]. These results support the above in vitro studies that G4s largely remain formed and stabilized upon oxidation. A caveat of note is that the BG4 antibody used by Miclot, et al. only detects formed G4s, so partial or destabilized G4 structures are not identifiable. Given that chronic inflammation is a hallmark of cancer, it will be particularly important to determine if similar results are obtained in cancer cell lines and with various oxidants. 

### 3.2. Base Oxidation in Loop Sequences

Though guanine is the most susceptible base to oxidation, it would be remiss to not discuss oxidation of the other three bases that can occur at G4 loop sequences. Loop sequences contribute to formation and stability of the G4, so it is necessary to understand oxidative damage at these sites. In the same way that a hydroxyl radical can lead to an 8oxoG, 8-oxo-7,8-dihydroadenosine (8oxoA) is a major product of adenine oxidation [76]. Interestingly, the effects of the 8oxoA lesion are also dependent upon G4 location (i.e., which loop sequence harbors the lesion). PAGE analysis and CD spectra of the 22-nucleotide oligo representing the telomeric G4 (AG_3_(TTAG_3_)_3_) in the presence of Na^+^ or K^+^ showed presence of the adduct did not substantially alter the antiparallel G4 structure [86]. Additionally, thermal melting showed slight variability between sequences harboring different lesions. The 8oxoA at the first loop or third loop induced a marginal increase in thermostability, suggesting a slight increase in G4 stability. Again, there is a dependence on the position of the 8oxoA, albeit only mild effects were observed. Apurinic (AP) sites at the same positions in the loops were investigated as well, showing a stronger dependence on location. For instance, the CD spectrum of the AP site located in the third loop changed compared to undamaged DNA [86]. This also suggests that the absence of adenine rather than just replacement with an oxidation product impacts G4 conformation, whether it is the structure itself or H-bonding partners. 

One of the main products of thymidine oxidation is 5-hmU [76]. As with 8oxoA, the effect of 5-hmU in place of thymidine at individual loop sequences has been studied in vitro. Biochemical analyses have shown 5-hmU has inconsequential effects on G4 conformation and stability [76,87]. Interestingly, these authors suggested that 5-hmU at loop positions created more inflexible secondary structures [87]. Other thymidine oxidation products exist, and it will be important to assess those lesions, especially in the context of other (non-telomeric) G4 motifs, particularly those with varying loop lengths. Short loop lengths increase the stability of the G4, and long loop lengths decrease its stability. The telomeric G4 has loop lengths of three nucleotides, so oxidation of G4 motifs with longer loop lengths may have a negligible effect, whereas oxidation of G4 motifs with a single nucleotide loop may have a more substantial impact.

To our knowledge, cytosine oxidation within G4 loops has not been extensively investigated. However, one study from Morgan, et al. exploring the impact of 5-hydroxymethylcytosine (5-hmC) on the G4 within the *VEGF* promoter may provide insight. In general, G4 oligonucleotides containing a 5-hmC modification at different loop locations did not disrupt G4 formation, according to circular dichroism analysis [88]. *VEGF* G4 wildtype and 5-hmC containing oligonucleotides were also incubated with the G4 stabilizer TMPyP4 or the protein nucleolin, whose ability to bind G4s has been established. Overall, G4s containing 5-hmC could not be efficiently stabilized by TMPyP4 and nucleolin exhibited reduced or no binding to G4s containing 5-hmC, indicating the correct nucleotide composition is important for stabilizer and protein binding [88]. However, other G4 stabilizers and proteins known to bind to G4s should be investigated to determine if this is a common trend. Additionally, there may be a dependency on sequence and/or conformation, as well as the type of cytosine modification.

Although limited, the available evidence suggests that oxidation at loop sequences has less of an impact on G4 stability and conformation than does oxidation at G-tracts. Translated to cells, it is possible that oxidation at loop sequences may be favorable in maintaining conformation and/or stability, depending on location and function. 

### 3.3. DNA Polymerase Response to Oxidative Lesions at Guanines

If DNA lesions, like 8oxoG, are not repaired by the cell (e.g., through base excision repair) they will be substrates for DNA replication. Given that polymerase bypass of 8oxoG has been well-studied, we will focus on the response to 8oxoG by polymerases implicated in G4 synthesis. For further discussion on polymerase response to other oxidative lesions, see the review from Berquist and Wilson [89]. 

During times of oxidative stress, Y-family polymerases appear critical in preserving genome integrity. From in vitro studies, replicative yeast pol δ has difficulty replicating and extending opposite 8oxoG lesions, showing very low bypass efficiency compared to templates containing an undamaged G at the same position [90,91]. Human pol δ (hpol δ) also has compromised accuracy and efficiency when bypassing 8oxoG [91]. Therefore, bypass by TLS polymerases may be more important for maintaining genome integrity at oxidized G4 motifs. 

#### 3.3.1. Pol η

Pol η is a likely player in the successful and efficient bypass of 8oxoG lesions. Yeast pol η can efficiently bypass 8oxoG and extend past the lesion [90,91], as evidenced by in vitro steady-state kinetics analyses. Though yeast pol η is more accurate than mouse and hpol η, hpol η can still synthesize past this lesion with more ease than other Y-family polymerases [91]. Several biochemical studies using the catalytic core of hpol η has shown pol η can efficiently and accurately insert C opposite the 8oxoG lesion and does so with higher efficiency than mispairing with A [90,92,93]. Comparisons of Y-family pols hpols η, κ, iota (ι), and hRev1 to hpol η alone suggest that pol η may be the major enzyme necessary for 8oxoG bypass [93]. 

#### 3.3.2. Pol k

Less evidence suggests a role of pol κ in bypassing 8oxoG. In vitro, hpol κ stalls opposite 8oxoG, and when extending [93]. This agrees with steady-state kinetics analysis of hpol κ, revealing that hpol κ has substantially reduced efficiency when inserting C opposite 8oxoG when compared to templates containing an undamaged G at the same position [94]. Importantly, hpol κ frequently and efficiently misincorporates A opposite 8oxoG in vitro, as evidenced by steady-state kinetics [94] and a short oligonucleotide sequencing assay [93]. By itself, hpol κ does not seem like a feasible candidate recruited for replication at guanines under oxidative stress. However, hpol κ still may play a crucial part in suppressing mutagenesis under oxidative stress conditions. Hpol κ may possibly act in conjunction with another polymerase. In vitro, hpol δ together with hpol κ can insert and extend from 8oxoG more efficiently than hpol κ by itself [94]. Hakura, et al. treated pol κ^−/−^ and pol κ^+/+^ mice with either the mutagen benzo(a)pyrene (BP), DSS, or both. There was no significant difference in the incidence of tumor development between pol κ^−/−^ and pol κ^+/+^ mice when treated with both BP and DSS, suggesting that pol κ does not function to repress tumorigenesis [95]. However, using a *gpt* transgene reporter, the authors showed that DSS treatment alone increased mutagenesis in pol κ^−/−^ mice, and not in pol κ^+/+^ mice, with G > C transversions being the most frequent errors in both the distal colon and lungs of these mice [95]. Taken together, pol κ may display lower fidelity during 8oxoG TLS, but the described in vivo study suggests it does play a role in suppressing mutagenesis in an oxidative stress environment.

#### 3.3.3. Pol ι

Pol ι has not been thoroughly investigated for any possible role in G4 bypass; however, that does not rule out its possible recruitment at sites of G4s during oxidative stress. Indeed, pol ι can insert C opposite 8oxoG most of the time but is negatively affected in its ability to correctly incorporate other bases in vitro [93]. Additionally, pol ι was previously implicated in base excision repair in SV40-transformed human lung fibroblasts (MRC5-SV) following H_2_O_2_-induced oxidative damage [96]. If oxidative damage at G-tracts is replicated, pol ι could be recruited to these sites or may be a critical replicative enzyme upon repair of this damage.

#### 3.3.4. Rev1 

Current evidence suggests that complete replication of G4s may entail recruitment of Rev1 under normal physiological conditions. Whether this is true of Rev1 when cells are under oxidative stress remains unclear. Rev1 and pol κ proved to be an efficient combination in vitro for 8oxoG bypass, as C was correctly inserted opposite 8oxoG ~92% of the time compared to A misinsertion; however, a significant number of single nucleotide deletions at the flanking nucleotides was observed [93]. These data suggest that Rev1 could function to correctly insert C opposite 8oxoG and act as a scaffold to recruit another polymerase that efficiently extends the lesion base pair. 

In summary, the terminal G-tetrads and loop sequences are more susceptible to oxidation than central G-tetrads, suggesting a possible protective effect that allows for continued stabilization of the G4. Given the implied roles of specialized polymerases in facilitating both the bypass of G4s and 8oxoG lesions, the recruitment of these enzymes to such sites during times of oxidative stress is a question that deserves investigation. G4 oxidation presents a two-fold issue for a polymerase to overcome during replication: (1) secondary structure and (2) oxidative lesion(s). The secondary structure would need to be resolved or bypassed, which was described in the previous sections. If the oxidative lesion is not excised and repaired, then a Y-family polymerase is likely to be recruited to bypass the lesion and maintain genome integrity. What would determine which Y-family polymerase(s) are recruited to G4s under oxidative stress? Our knowledge of G4s does not point to a clear answer, but evidence indicates that this could vary depending on G4 motif sequence and conformation. For instance, if polymerase recruitment depends on G4 conformation, then oxidative lesions that change the structure even slightly may change which polymerase is recruited. Additionally, a lesion like 8oxoG can be accurately synthesized in many cases by a specialized polymerase like pol η; however, these are low fidelity polymerases and errors will occur. The presence of an oxidative lesion within a G4 may facilitate mutagenesis. In some cases, inaccurate synthesis may lead to a mutated G4 motif that may have altered conformation, which may have functional consequences depending on the G4’s genomic location. 

## 4. Potential Indirect Impacts of Inflammation on G4 Stability

Chronic inflammation promotes profound changes to the tumor microenvironment that contribute to neoplastic progression. Our knowledge of the mechanisms by which stromal-epithelial interactions contribute to chronic inflammation-induced genome stability has been examined from the perspective of RONS and cytokines produced by inflammatory cells. For instance, myeloid cell-derived ROS causes increased DNA damage and mutagenesis in intestinal epithelial cells and contributes to tumorigenesis by inducing epithelial cell secretion of cytokines in a feed-forward loop [97]. Below, we highlight another potential RONS target that could alter gene expression; namely, G4 motifs. 

### 4.1. 8oxoG Formation at G4s in Promoters Regulate Gene Expression

PG4 motifs of high thermostability are overrepresented within functional genic components (e.g., promoters, 3′ and 5′ UTRs) and subject to purifying selection, consistent with a regulatory role for G4s in gene expression [31]. Studies have identified G4s in the promoter regions of oncogenes (e.g., c-*myc*, H-*ras*, see ref. [98] for further review), cancer-related genes (e.g., *VEGF* [98,99]), and genes related to development and neurologic function in humans [30]. As discussed above, oxidation at such G4s may or may not have detrimental effects on that regulation, depending on the location of the lesion within the G4. Those lesions that minimally affect G4 stability would allow for continued G4 formation and usual gene expression control. 

A critical factor determining whether oxidation affects G4 stability includes the number of G-tracts, as motifs with a fifth G-tract can stabilize the structure when the central G-tract in the motif is oxidized [100]. However, there is more of a consequence to this than just stabilizing the secondary structure. One study that investigated the effects of 8oxoG on gene expression by incorporating the *VEGF* promoter G4 motif with 8oxoG lesions into a luciferase reporter highlighted downstream regulation of expression [101]. Briefly, elevated Renilla luciferase was observed independent of 8oxoG position but was largely unchanged in OGG1^−/−^ MEFs compared to OGG1^−/−^ MEFs without 8oxoG containing plasmids. It was suggested that 8oxoG, excised by OGG1, generates an AP site, and new H-bonding and ion coordination with the fifth G-tract allows for APE1 binding and an increase in transcription [101]. Other impacts of oxidative damage on G4s involved in transcription regulation, including stabilization that decreases gene expression, are possible and not thoroughly discussed here. However, oxidative damage to G4s is clearly an additional challenge in transcription regulation, particularly the regulation of cancer-related genes. 

### 4.2. Potential Impact of G4s Located in Genes Related to Inflammation

Research investigating the direct and/or indirect impact of G4s within genes related to the inflammatory response is limited. A recent study by Stefan Bidula lays a foundation for this area of study by analyzing genes from the Eukaryotic Promoter Database and using G4Hunter to identify PG4s [102]. PG4 frequency within the promoters of immune-related gene families was variable in comparison to PG4 frequencies in oncogene promoters. Of note, genic PG4s were identified in several interleukin (IL), colony stimulating factor (CSF), CC chemokine (CCL), and tumor necrosis factor (TNF) gene families [102]. Whether any of these G4s are biologically relevant remains to be determined. G4s in the promoters of these genes may have a similar regulatory function to G4s in the promoters of oncogenes, where the formation of and/or protein recruitment to the G4 acts as an on/off switch for transcription. In the context of chronic inflammation, G4s may in this way contribute to the dysregulation of cytokines and chemokines at the transcriptional level. 

## 5. Discussion: G4s within the Context of Chronic Inflammatory Diseases

Here, we provide a foundation for the interplay between G4s and chronic inflammation. G4s may create an additive effect contributing to dysfunctional replication and/or transcription, among other processes not discussed. Consequences may be wide-ranging, necessarily increasing genome instability in the context of persistent, inflammation-related injury. Several questions have been posed throughout this review related to DNA polymerase engagement at, and synthesis through, G4s upon oxidative damage. Mechanistic insights into the susceptibility of G4s to oxidation and elucidation of novel replication processes at these sites to maintain genome stability are critical. Notwithstanding helicase involvement, recruitment of specialized polymerases in vivo may be necessary to overcome not only G4s but also for oxidative damage translesion synthesis at these sites. We note that consequences to G4 stability depend on sequence, type and location of lesion, which also may influence polymerase recruitment. Regardless, error-prone synthesis at these sites may increase mutational burden, thereby increasing genome instability. Additionally, failure to efficiently bypass oxidatively damaged G4s would lead to DNA breaks and structural variation.

## Figures and Tables

**Figure 1 genes-12-01779-f001:**
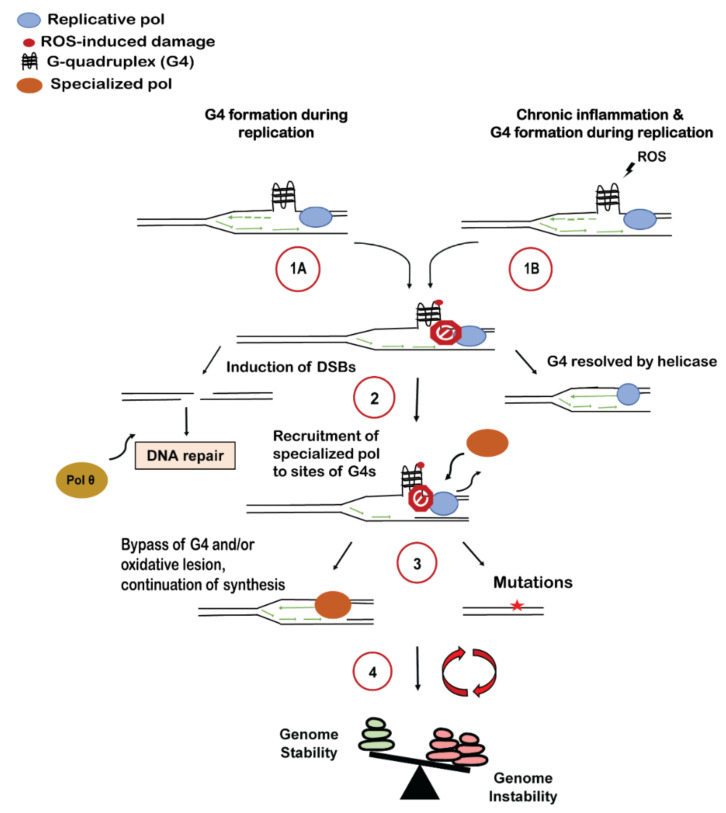
Consequences of G4s on DNA Replication within the Context of Chronic Inflammation. Several questions are posed throughout this review related to DNA polymerase engagement at, and synthesis through G4s, upon oxidative damage, and are summarized in this schematic. (**1A**) G4s can form during replication under normal, physiological conditions. A subset of G4 motifs can lead to aberrant replication fork progression, due to inhibitory effects of structure formation on polymerase activity alone and/or in conjunction with protein recruitment to regulate replication. (**1B**) Oxidative stress from repeated bouts of inflammation causes persistent ROS-induced DNA damage, and G4s are particularly susceptible sequences. This creates a dual challenge during replication: synthesis of undamaged G4s and bypass of oxidative damage at G4s. (**2**) Stalled forks caused by G4s (undamaged or damaged) have several fates: fork breakage and subsequent DNA repair (e.g., end-joining through pol θ); structure unwinding by a specialized DNA helicase; or recruitment of specialized DNA polymerases for completion of synthesis at G4s (including post-replicative gap-filling). (**3**) Recruitment of specialized polymerases to bypass the G4 and/or oxidative damage results in continued synthesis and mutagenesis. (**4**) Repeated oxidative damage with unrepaired double-strand breaks and recruitment of error-prone polymerases leads to increased genome instability.

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
