# Peer review of "Impact of G-Quadruplexes and Chronic Inflammation on Genome Instability: Additive Effects during Carcinogenesis"

_genes, 2021, doi:10.3390/genes12111779_

Round 1

Reviewer 1 Report

The review article from Stein and Eckert regarding the interplay of G4s and inflammation is thorough and intriguing.  Overall the authors encompass an array of research and tie together the fields nicely. There are a few minor issues that, when resolved, would improve the readability of the article to a broad audience, but overall the review article is well done and will be well received by the target audience.

Minor suggestions:

There are a few times where an acronym or a concept/drug are introduced with no explanation. For example, line 194 mentions aphidicolin, but do not mention what it is, and line 414 mentions Sp, but without explanation.

The use of in vitro and in vivo, to a broad audience, isn't likely to be clear.  Chemists use those terms to mean in a biological system versus in cells, whereas molecular biologist, pharmacologists, etc, use them to mean in cells and in animals.  These would lead to dramatically different conclusions of the results.  I suggest using more ubiquitous terms, such as "cell free" and "cellular" or "in cell", so that the audience is all on the same page.

Some of the references work, but are strange choices, such as reference 50 for identifying G4s in MYC (you could use the primary literature describing the various isoforms and conditions, or a review article - there are several to choose from) and 95 for KRAS (again versus using the primary literature for the two promoter regions). Adding the primary literature would is suggested.

Line 613 has an unformatted reference.

Lastly, the discussion on base oxidation in loop sequences (lines 482-515) doesn't discuss cytosines and it is a noticeable absence. I suggest adding text regarding modification of cytosines, or mentioning that work has not been done on them, whichever is the case. 

Author Response

We thank the reviewer for carefully assessing our manuscript and providing valuable feedback. Below are the point-by-point responses.

  1. Line 194 referencing aphidicolin is now accurately defined as a replication stress inducer that inhibits replicative polymerases and includes a reference for readers. Sp mentioned in line 414 was already defined in lines 409-410, and may have been missed by the reviewer.
  2. We appreciate the recommendations for use of in vitro and in vivo to a broader audience. We reviewed and edited the document to more clearly define in vitro or in vivo where use may have been confusing previously. In line 128, in vivo is defined in parentheses as “in cells or tissues”. In lines 217-218, the explanation of in vitro experiments from the Usdin lab now states the use of “biochemical assays with purified proteins” as the definition for in vitro. Additionally, for proteins or oligonucleotides used in certain in vitro experiments, we now describe them as “purified”, as in lines 217, 303, and 411. The terms “ex vivo” (lines 327, 361) or “in cells” (line 470) now are used to describe cell culture experiments where it may have been unclear. We removed a possibly confusing sentence (lines 554-555) that uses both in vitro and in vivo terminology. The main point of the sentence is technically reiterated in lines 556-557. Lastly, we removed in vivo from line 568, which may have been confusing and does not improve the meaning of the statement.
  3. The reviewer makes a fair point about the references used in lines 636 for myc and ras. As we used these only as examples of well-characterized oncogenes containing G4s in their promoters and did not expand upon them further, we now refer readers to a recent review (ref. 98) that discusses the G4s in these genes. We also include updated references for VEGF in line 637, ref. 98 and a primary article that describes the characterization of the G4 within the VEGF promoter.
  4. We thank the reviewer for catching the unformatted reference in line 613 (now line 629). It is now correctly formatted.
  5. We fully appreciate the point made by the reviewer on the lack of discussion of cytosine oxidation in G4 loops. Literature support is limited for this; however, we now include discussion of one study on the 5-hmC modification and the G4 in the VEGF promoter as representation for oxidative damage that may occur at cytosines in G4 loops.

Reviewer 2 Report

The manuscript “Impact of G-Quadruplexes and Chronic Inflammation on Genome Instability: Additive Effects During Carcinogenesis” discusses about the potential additive effects on cancer development of G4 formation plus G4 oxidation due to chronic inflammation. I have found the manuscript very well written and structured as well as comprehensive. The manuscript would be interesting to the field.

Minor point: Line 613, reference is not formatted properly.

Author Response

We thank the reviewer for carefully assessing our manuscript and for catching the unformatted reference in line 613 (now line 629). It is now correctly formatted.